# Differences in structure and hibernation mechanism highlight diversification of the microsporidian ribosome

Kai Ehrenbolger[1☉], Nathan Jespersen[1☉], Himanshu Sharma[1☉], Yuliya Y. Sokolova[2,3], Yuri S. Tokarev[4], Charles R. Vossbrinck[5], Jonas Barandun[1]*

1 Department of Molecular Biology, Laboratory for Molecular Infection Medicine Sweden, Umeå Centre for Microbial Research, Science for Life Laboratory, Umeå University, Umeå, Sweden, 2 Department of Microbiology, Immunology, and Tropical Medicine, School of Medicine and Health Sciences, George Washington University, Washington, District of Columbia, United States of America, 3 Institute of Cytology, St. Petersburg, Russia, 4 All-Russian Institute of Plant Protection, St. Petersburg, Russia, 5 Department of Environmental Science, Connecticut Agricultural Experiment Station, New Haven, Connecticut, United States of America

☉ These authors contributed equally to this work.
* jonas.barandun@umu.se

**Data Availability Statement:** The cryo-EM density maps for the microsporidian ribosome have been deposited in the EM Data Bank with accession code

## Abstract

Assembling and powering ribosomes are energy-intensive processes requiring fine-tuned cellular control mechanisms. In organisms operating under strict nutrient limitations, such as pathogenic microsporidia, conservation of energy via ribosomal hibernation and recycling is critical. The mechanisms by which hibernation is achieved in microsporidia, however, remain poorly understood. Here, we present the cryo–electron microscopy structure of the ribosome from *Paranosema locustae* spores, bound by the conserved eukaryotic hibernation and recycling factor Lso2. The microsporidian Lso2 homolog adopts a V-shaped conformation to bridge the mRNA decoding site and the large subunit tRNA binding sites, providing a reversible ribosome inactivation mechanism. Although microsporidian ribosomes are highly compacted, the *P. locustae* ribosome retains several rRNA segments absent in other microsporidia, and represents an intermediate state of rRNA reduction. In one case, the near complete reduction of an expansion segment has resulted in a single bound nucleotide, which may act as an architectural co-factor to stabilize a protein–protein interface. The presented structure highlights the reductive evolution in these emerging pathogens and sheds light on a conserved mechanism for eukaryotic ribosome hibernation.

## Introduction

Ribosome biogenesis and functionality are energy-intensive processes accounting for 80% of a cell's ATP usage in nutrient-rich conditions [1,2]. When nutrients are scarce, many cells transition to a dormant state typified by low metabolic activity [3]. In this state, energy is conserved by inhibiting ribosomes via a diverse family of proteins known as hibernation factors [4]. These factors play an essential role in both the inactivation of ribosomes and their recovery post-dormancy [5–7]. Several novel hibernation factors have been recently identified [7–9],

EMD-11437 (state 2, composite multibody refined map), EMD-11437-additional map 1 (LSU focused), EMD-11437-additional map 2 (SSU-body focused) and EMD-11437-additional map 3 (SSU-head focused). Coordinates have been deposited in the Protein Data Bank under accession code PDB-6ZU5.

**Funding:** N.J. is supported by an Integrated Structural Biology fellowship from Kempe and H.S. is supported by an individual Kempe fellowship (www.kempe.com). J.B. acknowledges funding from the Swedish Research council (2019-02011, www.vr.se), the SciLifeLab National Fellows program and MIMS. C.R.V. acknowledges funding from the Hatch Grant Project CONH00786 and R. Tyler Huning. Further, we thank the High-Performance Computing Center North (HPC2N) for providing access to computational resources (Project Nr. SNIC 2020/9-83 and SNIC 2020/10-83). The funders had no role in study design, data collection and analysis, decision to publish, or preparation of the manuscript.

**Competing interests:** The authors have declared that no competing interests exist.

**Abbreviations:** A-site, acceptor site; cryo-EM, cryo–electron microscopy; CTF, contrast transfer function; E-site, exit site; EM, electron microscopy; ES, expansion segment; LSU, large subunit; P-site, peptidyl site; SSU, small subunit.

including the conserved microsporidian dormancy factor Mdf1 and the species-specific factor Mdf2 [9].

Microsporidia are obligate intracellular parasites that infect organisms as evolutionarily divergent as protists and mammals [10]. Widespread microsporidian infections have particularly deleterious effects on immunodeficient patients [11] and commercially important silkworms and honeybees [12]. Despite their perniciousness, recent work has shown that certain microsporidia can impair the transmission of *Plasmodium falciparum*, the causative agent of malaria, thereby providing a potential mechanism to control this highly infectious human pathogen [13].

The microsporidian life cycle alternates between multiple proliferative intracellular stages and a sporous, metabolically inactive environmental stage [10]. Due to extreme genome compaction and loss of seemingly essential pathways such as nucleotide synthesis [14] and ATP generation via oxidative phosphorylation [15], the intracellular stages depend heavily on host cells for their energy requirements [14]. In the spore stage, the limited availability of nutrients and the requirement for rapid reactivation of essential cellular processes after host infection necessitate efficient reversible hibernation mechanisms.

The extreme genome compaction in microsporidia, which has resulted in the smallest known eukaryotic genomes [16,17], has affected not only metabolic genes but also ancestral macromolecules like the ribosome [9]. In most eukaryotes, ribosomes contain expanded rRNA elements, termed expansion segments (ESs), and numerous eukaryotic-specific proteins [18]. While most of these ESs stabilize the additional layer of proteins [19], it is suggested that some aid in ribosome biogenesis [20], and others extend the functional repertoire of the ribosome by providing additional interaction interfaces for regulatory factors [21]. Interestingly, microsporidia have reversed this evolutionary expansion and removed previously acquired elements as a part of their continued genomic compaction [9,22]. In the extreme case of *Vairimorpha necatrix*, this has created one of the smallest eukaryotic cytoplasmic ribosomes [9]. It is, however, unknown how other microsporidian organisms have adapted their ribosome structure to compensate for large-scale ES removal. Therefore, microsporidia are ideal model organisms to study rRNA evolution, as well as ribosomal hibernation due to their conspicuous dormancy.

Here, we use cryo–electron microscopy (cryo-EM) to solve the structure of the ribosome from *Paranosema locustae*, a species of microsporidium used as a microbial insecticide to control grasshopper and locust pests [23,24]. A comparative analysis of the ribosomal ESs present in *P. locustae* with the related *Saccharomyces cerevisiae* (yeast) and *V. necatrix* (microsporidium) structures gives insights into the reductive evolution of the ribosome in microsporidia. A single structural nucleotide, discovered at the interface of 2 ribosomal proteins, serves as the remaining element of a removed rRNA segment and may act as the most minimal version of an ES. Furthermore, we identify a non-ribosomal protein bound to the *P. locustae* spore ribosome as Lso2 (late-annotated short open reading frame 2), a recently discovered hibernation and recovery factor [8]. We present the first structural description of this factor in microsporidia and propose a conserved functional role in other eukaryotic organisms. Together, these results provide insights into the reductive nature of microsporidian evolution and unravel a novel mechanism of translational shutdown in the extracellular stage of these emerging pathogens.

## Results

### The cryo-EM structure of the ribosome from *P. locustae*

To study the microsporidian ribosome and its interaction partners during the dormant extracellular stage, we isolated ribosomes from *P. locustae* spores and analyzed them using cryo-EM

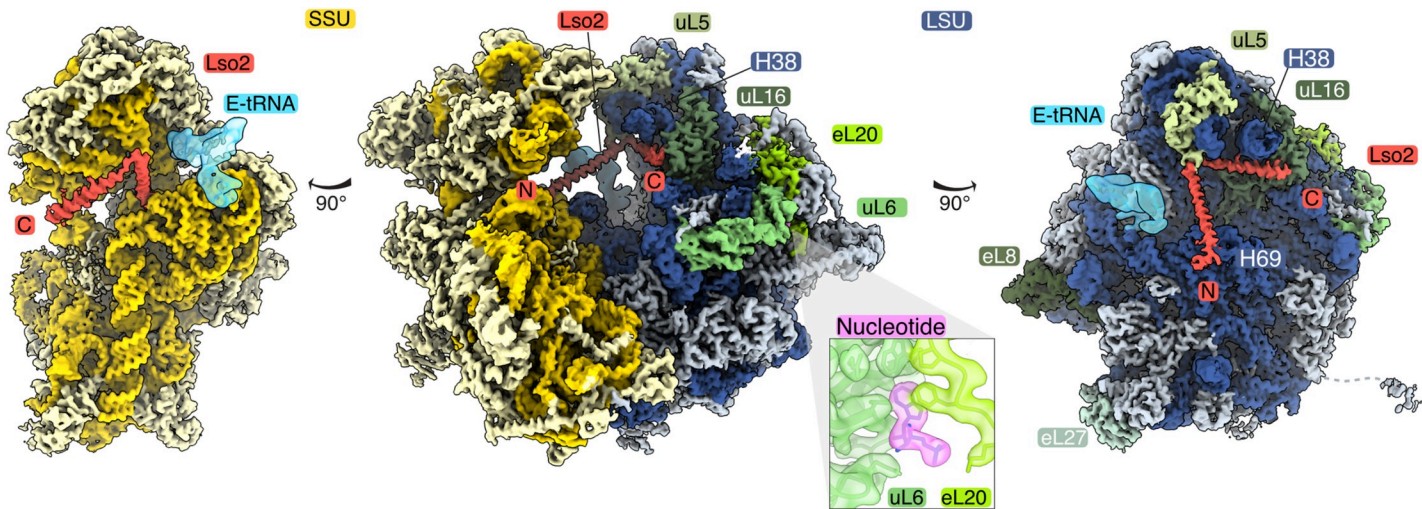

**Fig 1. The cryo-EM structure of the *P. locustae* ribosome, bound to Lso2 and a structural nucleotide.** Composite cryo-EM map consisting of maps focused on the LSU, SSU-body, and SSU-head is shown (EMD-11437). The complete ribosome is shown in the center, while the SSU (left) and LSU (right) are depicted in isolation on both sides. The SSU is colored in shades of yellow (RNA in gold, proteins in light yellow), while the LSU is colored in shades of blue (RNA in dark blue, proteins in light blue), with selected ribosomal proteins labeled and colored in shades of green. The hibernation and recovery factor Lso2 is highlighted in red. The inset showcases the nucleotide-binding site (purple) at the interface between the 2 LSU proteins uL6 and eL20 (shades of green), displayed by superimposing the cryo-EM map with the molecular model. C, C-terminus; cryo-EM, cryo–electron microscopy; E-tRNA, exit site tRNA; LSU, large subunit; N, N-terminus; SSU, small subunit.

(Figs 1 and S1). A consensus refinement resulted in a cryo-EM map at an overall resolution of 2.7 Å, with a well-resolved large subunit (LSU) and a dynamic and less-resolved small subunit (SSU)–head region. A 3D classification focused on the mobile SSU-head was performed to improve this region, resulting in 2 states with either a rotated (State 1, 37.7%) or non-rotated (State 2, 39.6%) conformation (S1B Fig). The non-rotated State 2 contains additional, but poorly resolved, density for an exit site (E-site) tRNA (Fig 1). Multibody refinement of State 2 improved the local resolution for the LSU (2.83 Å), the SSU-body (3.03 Å), and the SSU-head (3.27 Å) (S1B and S2 Figs). The improved resolution allowed for model building of the *P. locustae* State 2 ribosome structure, using the *S. cerevisiae* ribosome as a starting template [19] (S1 and S2 Tables). The L1 stalk, L7/L10 stalk, and parts of the SSU-beak were not resolved and therefore not included in the final model. We observed an additional acceptor site (A-site)/peptidyl site (P-site) helical density, spanning from the SSU to the LSU central protuberance (Fig 1). The high occupancy (92%; S1B Fig) and resolution of this density allowed for its unambiguous assignment as the eukaryotic hibernation and recovery factor Lso2 [8]. Both conformations of the SSU-head contain Lso2 density, suggesting it neither stabilizes one particular state nor binds in concert with the E-site tRNA. In addition, we discovered density for a nucleotide with a local resolution of 3 Å in the interface between uL6 and eL20 (Figs 1 and S2D), acting as a remnant of a removed ES.

## The ribosome hibernation and recovery factor Lso2 blocks key catalytic sites

The microsporidian homolog of Lso2 is bound to the central cavity of the *P. locustae* ribosome (Figs 1 and 2). In yeast, Lso2 has been recently identified as a hibernation and recovery factor that binds to ribosomes during starvation [8]. A BLAST search allowed us to verify the presence of Lso2 in almost all sequenced microsporidia (S3A Fig). In *P. locustae*, Lso2 is a 77-amino-acid protein comprising 2 consecutive alpha helices (α1, α2), separated by a small hinge region (Figs 2 and S3A). Residues 2–76 are well resolved and included in our model.

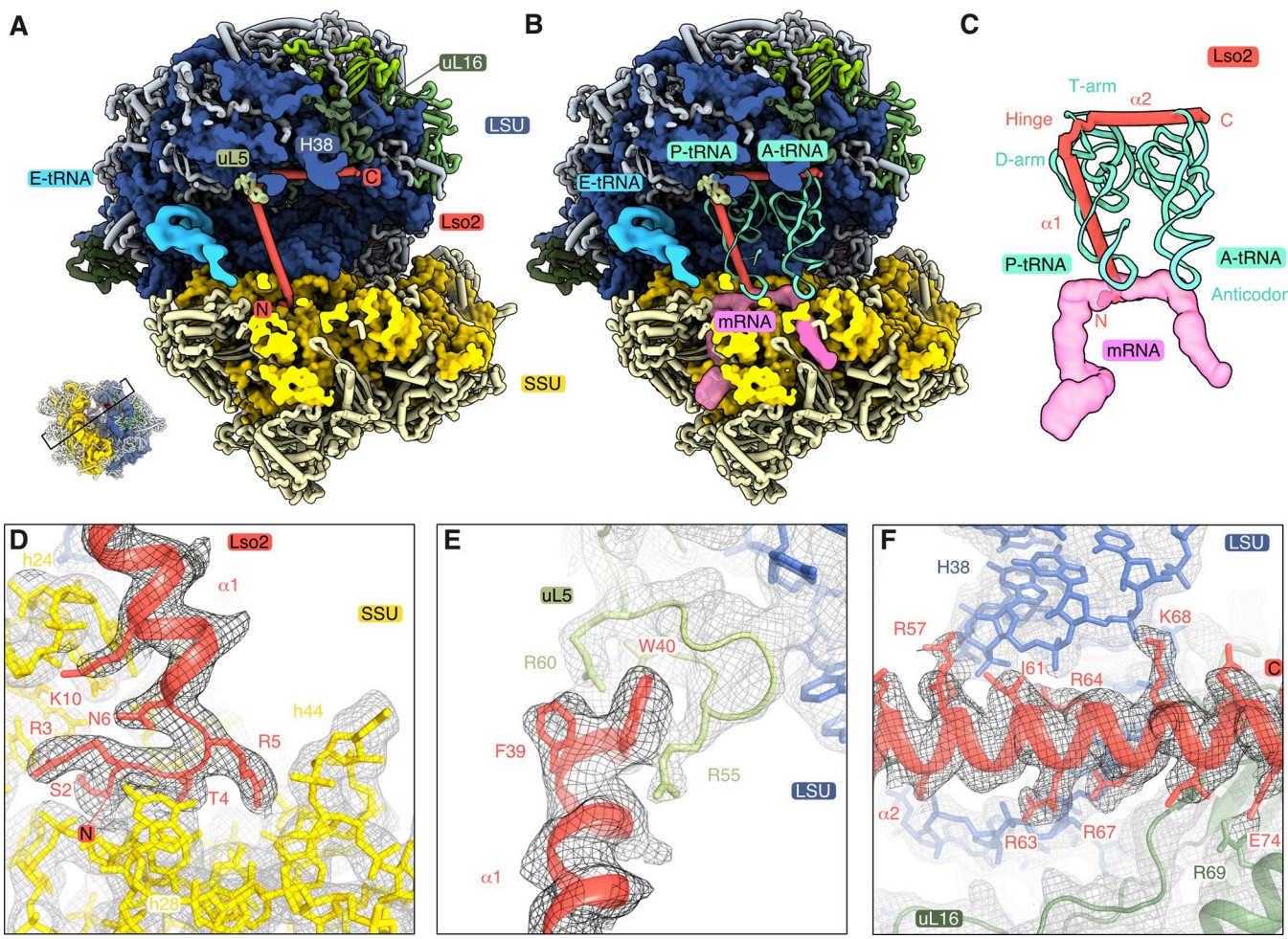

**Fig 2. Microsporidian Lso2 interactions with the ribosome.** (A) Slab view of Lso2 (red) bound ribosomes along with the cryo-EM density for E-site tRNA (sky blue). Lso2 ends contacting the SSU and LSU are indicated as N and C, respectively (PDB 6ZU5). (B) Lso2 prevents tRNA and mRNA binding in the A- and P- site as shown by the superimposed tRNAs (aquamarine, from PDB 4V6F) and an mRNA (pink surface, from PDB 4V6F). (C) An isolated view of Lso2 sterically blocking the codon–anticodon site, as well as the D- and T-arm of the P-site tRNA. The C-terminal end overlaps with the T-arm of the A-site tRNA. (D–F) Molecular contacts between Lso2 and the ribosome, shown as cryo-EM density (mesh) and the structural model. Lso2 residues contacting the rRNA or ribosomal proteins are indicated. A-site, acceptor site; A-tRNA, acceptor site tRNA; C, C-terminus; cryo-EM, cryo–electron microscopy; E-site, exit site; E-tRNA, exit site tRNA; LSU, large subunit; N, N-terminus; P-site, peptidyl site; P-tRNA, peptidyl site tRNA; SSU, small subunit.

The N-terminal extension preceding α1 contacts the SSU mRNA binding channel between helices h24, h28, and h44 (Fig 2D). The first helix (α1) connects this binding site with ribosomal protein uL5 at the central protuberance of the LSU (Fig 2E). Several positively charged and conserved residues (K9, K10, K17) anchor the beginning of α1 to the decoding center in the SSU rRNA (Fig 2D). While spanning the central cavity, Lso2 anchors to the 25S rRNA backbone of helix-69 using R16, and stacks W40 between R55 and R60 from uL5 (Fig 2E). After a short hinge region, a second helix (α2) extends from the LSU P-site to the A-site by fitting into the major groove of H38A (Fig 2F). Here, a patch of conserved positively charged residues interact with the rRNA backbone and anchor the C-terminal end of α2 to uL16.

Lso2 blocks the binding sites of 3 essential components of the translational machinery. While the N-terminal extension occupies the mRNA channel in the SSU, α1 superimposes with the D-arm of the P-site tRNA, and α2 sterically blocks the T-arm of both P-site and A-site tRNAs (Fig 2B and 2C). Extensive binding site overlap supports the role of Lso2 as a

hibernation factor in microsporidia and indicates that its removal is required for reactivation of protein synthesis upon infection of a host. The general conservation of SSU- and LSU-interacting residues suggests that Lso2 would adopt a similar binding mechanism in other microsporidia as well as other eukaryotes (S3 Fig).

### A bound nucleotide as evidence for adaptation to ES loss

A comparison of the *P. locustae* ribosome to the yeast and *V. necatrix* structures (Fig 3) demonstrates that microsporidia commonly reduce protein size and remove ESs during genome compaction. Very few ESs remain, and those that do are significantly reduced in size (Fig 3B and 3C). A previous analysis of microsporidian SSU rRNA revealed that microsporidia are evolving towards rRNA reduction [9]. Early-branching species like *Mitosporidium daphniae* contain longer and more numerous ESs, while recently branched species have eliminated these sequences. *V. necatrix* has lost nearly all eukaryotic-specific ESs and several additional helices of the universal core [9]. In contrast, rRNA removal has not progressed to the same extent in *P. locustae*, and remnants of ancestral elements are still present. One such example is the functionally important region surrounding the polypeptide exit tunnel in the LSU, where H7, H19, and H24 share a high structural similarity with yeast and form a narrow channel (Figs 3 and S4A). Contrastingly, *V. necatrix* has removed these elements and features a broad and open tunnel exit. In the SSU, the 2 large ESs es6 and es3 are entirely absent in *V. necatrix*, while they are still partially present in the *P. locustae* structure (Figs 3 and S4B). A section of *P. locustae* es6 (es6C) again superimposes well with the yeast counterpart, whereas the short es6D and the 3 larger segments es6A, es6B, and es6E have been eliminated (S4B Fig). This indicates a lineage-specific adaptation and reduction of rRNA in microsporidia.

A notable example of adaptation to ES loss can be seen in the *P. locustae* structure, where the elimination of ESs may have resulted in poorly stabilized interactions between ribosomal proteins (Fig 4). In yeast and many other eukaryotic ribosomes, a nucleotide from ES39 (A3186 in yeast) is inserted into a crevasse between uL6 and eL20 (Fig 4A and 4C). Although the *P. locustae* rRNA does not contain this ES (Fig 4B), extra density between uL6 and eL20 is consistent with a free nucleotide (Figs 4D and S2D). This suggests that ES39 served a vital role in the stabilization of the protein–protein interface, and the nucleotide-binding site has been retained despite the loss of ES39. The *V. necatrix* ribosome, on the other hand, does not contain this nucleotide-binding site and is potentially demonstrative of a later evolutionary state post-ES loss (Fig 3C). Consistently, only some of the earliest diverging microsporidian species, like *M. daphniae*, encode ES39.

Although previous work has suggested that the loss of ESs necessitates a loss of protein binding sites on the ribosome [22], our data indicate that this is not the case. In particular, removal of the majority of the eL8 and eL27 ES binding sites does not lead, as proposed [22], to the loss of these 2 proteins. Both proteins are bound to the *P. locustae* ribosome at high occupancy (Fig 1), indicating that a small number of important and conserved interaction loci are sufficient for binding. On the other hand, the ribosomal proteins eL38 and eL41 of the LSU are absent in our *P. locustae* structure, as has been noted for *V. necatrix* [9], suggesting absence in most microsporidian species. Finally, no density was visible for the microsporidian-specific ribosomal protein msL1 in *P. locustae*.

## Discussion

Microsporidia are divergent pathogens that infect essentially all animal species [25]. Despite their broad host range and recognition as emerging pathogens of high priority [26], little is known about the evolution of their molecular machinery, which has been shaped by extreme

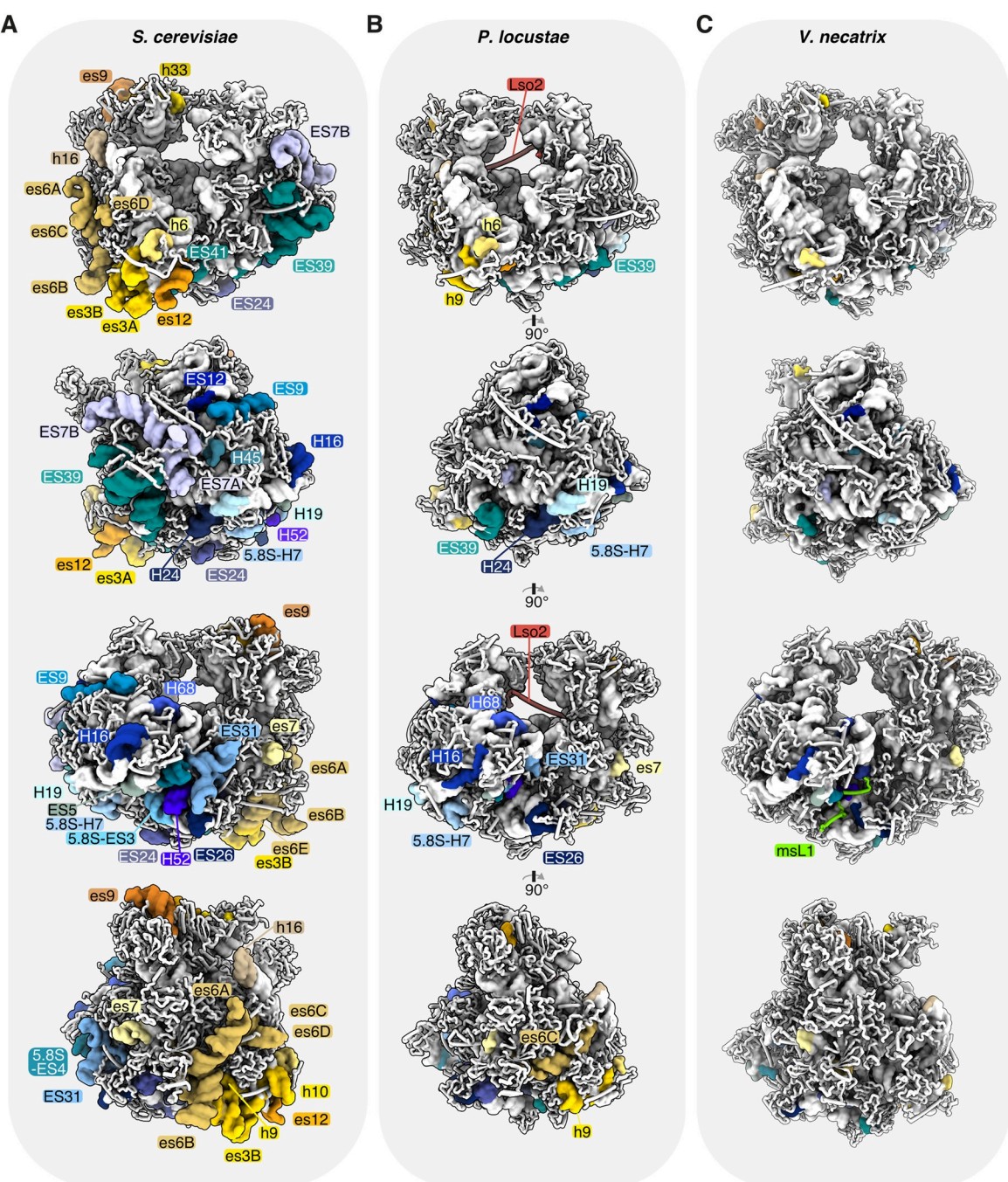

**Fig 3. Ribosomal RNA compaction in microsporidia.** Comparative analysis of expansion segments and the absence thereof between (A) *S. cerevisiae* (PDB 4V88), (B) *P. locustae* (PDB 6ZU5), and (C) *V. necatrix* (PDB 6RM3). (A–C) Four 90˚-related views of the ribosome are shown (from top to bottom), with expansion segments colored and labeled in shades of yellow and orange (small subunit) or shades of blue and green (large subunit).

genome compaction and an accelerated evolutionary rate [27]. Previous work has shown that the reductive nature of these pathogens has resulted in one of the most compacted eukaryotic cytosolic ribosomes [9]. The significant sequence divergence between microsporidian species suggests variability in microsporidian adaptation to genome compaction and nutrient

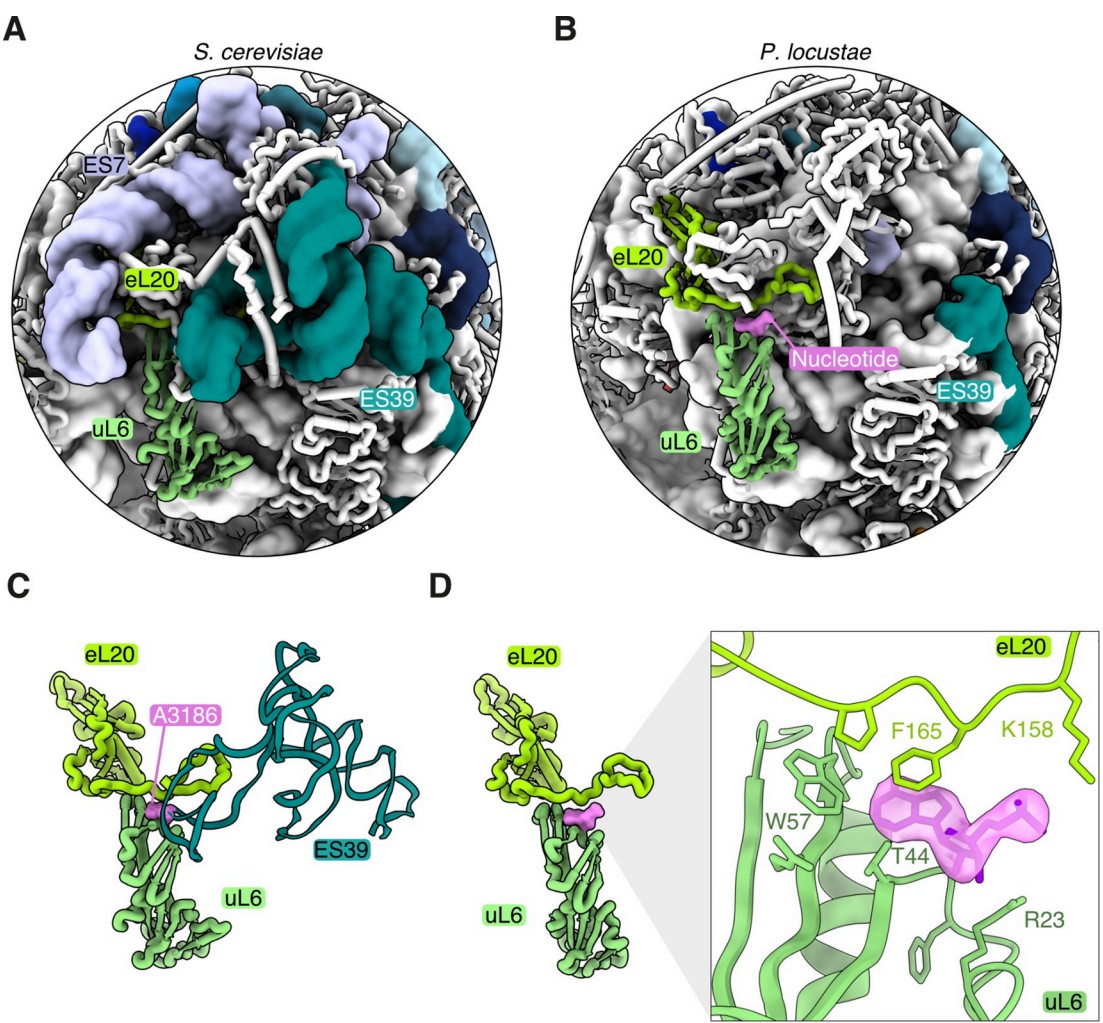

**Fig 4. Reductive evolution of ES39 to a single structural nucleotide.** A comparison of ES7 and ES39 between (A) *S. cerevisiae* (PDB 4V88) and (B) *P. locustae* (PDB 6ZU5). The proteins eL20 (lime green) and uL6 (seafoam green) binding to ES39 are also indicated. (C) An isolated, close-up view of the binding interface between eL20 and uL6, stabilized by A3186 (pink) from ES39 in the *S. cerevisiae* ribosome (PDB 4V88). (D) A single nucleotide (pink) is bound to the protein–protein interface in *P. locustae*. The inset depicts the cryo–electron microscopy map consistent with a single nucleotide between eL20 and uL6.

limitation. These differences can be visualized by comparing ribosome structure, composition, and hibernation mechanisms.

While most eukaryotic ribosomes contain extensive ESs to stabilize ribosome structure and facilitate interactions with various ribosome-associated proteins, a previous study on the microsporidian ribosome of *V. necatrix* has shown that nearly all of these ESs have been eliminated during genome compaction. In this study, we provide the first structural analysis of the *P. locustae* ribosome and demonstrate that it occupies an intermediate state of rRNA reduction between yeast and *V. necatrix*. One intriguing example of rRNA reduction is ES39, which is lost in both *V. necatrix* and *P. locustae*. In yeast, ES39 contacts several ribosomal proteins in the LSU by inserting a flipped-out base (A3186) into a binding site between uL6 and eL20. Interestingly, the deletion of ES39 in yeast is known to perturb ribosome assembly and leads to accumulation of assembly intermediates [20]. In the presented cryo-EM map, we observe clear density for a free nucleotide that superimposes well with yeast A3186 (Figs 4 and S2D). It is surprising that a nucleotide-binding site would be conserved after the ES was eliminated,

especially since no nucleotide density was visible in the *V. necatrix* structure. One explanation is that *V. necatrix* represents a later evolutionary state in rRNA compaction, and that alterations in uL6 and eL20 have rendered the nucleotide-binding site unnecessary. In this case, the bound nucleotide in *P. locustae* would be a relic evincing the comparatively recent evolutionary loss of ES39. Although the high conservation of this binding site in eukaryotes suggests an important and conserved function, it is possible that this interaction is a result of proximity and opportunity. Further studies, e.g., involving the deletion of this nucleotide from yeast ES39, would be necessary to verify the functional significance of this interaction.

Interestingly, although the specific functional role of most ESs is largely unknown, ES27 has been implicated in translation fidelity [21]. Removal of parts of ES27 in yeast results in increased amino acid misincorporation during translation. The lack of ES27 in microsporidia suggests that microsporidia either encode a separate means to ensure translational fidelity or that they can tolerate a more error-prone system. Previous work supports the latter hypothesis, and one notable study demonstrated that microsporidian translation is surprisingly editing-deficient [28]. Although some misincorporation was compellingly linked to incorrect loading by amino-acyl tRNA synthetases, we hypothesize that the elimination of ES27 contributes to the low fidelity of microsporidian translation.

Translation is an energy-intensive process, requiring the hydrolysis of an estimated 30 nucleotides for the biosynthesis and polymerization of each amino acid [29]. Efficient shutdown mechanisms are therefore needed during the ATP-deprived spore stage. Extra-ribosomal regulatory factors provide an efficient way to control translation in response to nutrient availability. Ribosome hibernation factors (also known as ribosome dormancy factors [9] or silencing factors [30]) are a class of proteins that bind to and inactivate ribosomes. They function via 3 main methods: (1) promotion of ribosome 100S dimers by joining SSUs in a head-to-head fashion, as seen with the hibernation promoting factor (HPF) [31,32]; (2) inhibition of 70/80S ribosome formation, as with ribosomal silencing factor A (RsfA) [30]; or (3) stabilization of 70S/80S ribosomes and blocking binding sites for various translational machinery, as with interferon-related developmental regulator 2 (IFRD2) [7]. Recently discovered hibernation factors in *V. necatrix* function via the third method, where the conserved factor Mdf1 occupies the E-site tRNA binding site in the SSU while a species-specific factor, Mdf2, blocks the polypeptide transferase center and exit tunnel [9]. In a similar fashion, Lso2 interferes with key binding sites in the translation apparatus (Fig 2B and 2C).

In yeast, Lso2 has been identified as a protein required for translation recovery after glucose starvation [8]. The purification of the *P. locustae* ribosome from mature dormant spores and the high occupancy of Lso2 in our structure suggest that the hibernation function is important in the extracellular spore stage of microsporidia. Similar to the other 2 identified microsporidian dormancy factors, Mdf1 and Mdf2 [9], the presence of Lso2 is incompatible with active translation (Fig 2B and 2C). Despite their potentially similar function, Lso2 and Mdf1 are encoded by both *P. locustae* and *V. necatrix*, suggesting different modes of ribosome hibernation are required. Based on an overlapping binding site on uL5, we speculate that only 1 of the 2 factors can bind at a time. It is also possible that Mdf1 or Lso2 is involved in removing the other factor from dormant ribosomes, i.e., hibernation and recovery functions are performed separately. Previous analyses indicate that *mdf1* transcription is increased towards the end of sporulation [9], suggesting Mdf1 activity is controlled by regulating protein concentration. Although no similar data are available for Lso2 in microsporidia, other hibernation factors are rapidly degraded upon return to nutrient-rich conditions [33,34], suggesting temporal regulation is common. Alternatively, Lso2 interactions with ribosomes may be mediated by post-translational methylations to the numerous lysine residues present throughout the protein, or by phosphorylation of the relatively conserved serine and threonine residues at the N-terminal

protein–rRNA binding interface (Figs 2 and S3). Further work is needed to segregate the functional roles for various hibernation factors, and to identify the mechanisms by which hibernation factors are regulated. This cryo-EM structure serves as a model for the efficient shutdown of a mechanistically complex macromolecular machine using a small protein, and sheds light on the reductive characteristics of a unique and emerging pathogen.

## Materials and methods

### Cultivation of *P. locustae* and isolation of spores

The microsporidium *P. locustae* was propagated in a constant laboratory culture of the migratory locust *Locusta migratoria* (Orthoptera: Acrididae). *P. locustae* spore suspensions were mixed with small pieces of fresh maize leaves and fed to third instar locust nymphs at an infection ratio of $10^6$ spores/nymph. Nymphs were starved for 24 hours before infection. After complete consumption of contaminated forage, nymphs were maintained at 30˚C in wooden cages with metal grids and provided constant light and fresh maize foliage. After 40–60 days, surviving insects were dissected. Swollen adipose tissue, tightly packed with spores, was homogenized in a glass vial with a Teflon pestle. The homogenates were filtered using a syringe plugged with cotton, centrifuged at 1,000*g* for 5–10 minutes, and washed with distilled water [35,36]. The final spore pellet was stored at 4˚C prior to experiments.

### Purification of the *P. locustae* ribosome

Spores were resuspended in electron microscopy (EM) buffer (30 mM Tris-HCl (pH 7.5), 25 mM KCl, 5 mM magnesium acetate, 1 mM DTT, 1 mM EDTA) in a 2-ml microcentrifuge tube. To liberate ribosomes, 0.5 g of 0.5-mm zirconia beads was added to the tube, and the spores were lysed by bead beating for 30 seconds. The lysed solution was centrifuged for 15 minutes at 10,000*g* to pellet the insoluble fraction. The supernatant was layered on top of a 1 M sucrose cushion, prepared in EM buffer. Ribosomes were then pelleted by centrifugation at 105,000*g* for 4 hours at 4˚C. After removing the supernatant, the pellet was resuspended in 20 μl of EM buffer. To determine purity and concentration, 5 μl was added to 95 μl of EM buffer, and absorption was measured between 240 and 300 nm.

### Cryo-EM grid preparation and data collection

Sample quality and homogeneity were analyzed by cryo-EM. A Quantifoil R 1.2/1.3 Cu 300 grid (Quantifoil Micro Tools, Prod. No. 658-300-CU) was glow-discharged for 30 seconds at 50 mA prior to the addition of a 3.5-μl ribosome sample ($A_{260}$ 4 mAU). Grids were then flash-frozen in liquid ethane using an FEI Vitrobot (Thermo Fisher Scientific) set to 100% humidity, 4˚C, blot force of −5, waiting time of 1 second, and blot time of 4.5 seconds. Micrographs were collected at the Umeå Core Facility for Electron Microscopy on a Titan Krios (Thermo Fisher Scientific) operated at 300 kV, equipped with a Gatan K2 BioQuantum direct electron detector. EPU (Thermo Fisher Scientific) was used for the automated data collection of a total of 5,332 movies with 40 frames at a total dose of 28.6 e$^-$/Å$^2$, a pixel size of 1.041 Å, and a defocus range between 0.7 and 2 μm (S1 Table).

### Cryo-EM image processing

MotionCor2 [37] was used for the initial movie alignment, drift correction, and dose-weighting. The contrast transfer function (CTF) was determined using CTFFIND-4.1.13 [38]. Micrographs with poor CTF fits or drift were removed after manual inspection, resulting in a total of 5,274 micrographs. Particle picking was performed with the deep learning object detection

system crYOLO [39] using a trained model based on 2,781 manually picked particles. A total of 318,301 particles were initially picked. After manual inspection of all coordinates and micrographs, 320,669 particles were extracted with a box size of 400 pixels (416 Å) and subjected to an initial 2D classification to remove remaining picking contaminants. Well-resolved 2D class averages (314,213 particles) were used alongside a 60-Å lowpass-filtered initial cryoS-PARC [40] model for refinement in RELION-3.1 [41].

Consensus refinement of all particles resulted in a map of 3.04 Å, which was further improved by per-particle CTF refinement to an overall resolution of 2.73 Å. Despite well-resolved density for Lso2, a focused classification with 3 classes suggests that approximately 92% of all particles are bound by this protein (S1B Fig). Weak density for an E-site tRNA was observed, and conformational heterogeneity in the SSU-body and head region resulted in less well-resolved SSU density. To further improve the density for the SSU-head region, a 3D classification focused on the SSU-head and E-site tRNA without image alignment was performed using 3 classes (S1B Fig). Two of these classes displayed an improved overall resolution for the SSU-head and tRNA site. Class 1 contained 37.7% (118,600 particles) and Class 2 contained 39.6% (124,947 particles) of the total particle mass. An overlay of both classes suggests that they adopt different rotational states (S1B Fig). The particles of Class 2 were selected and refined to an overall resolution of 2.93 Å (S2A Fig). To improve resolution of the distinct subdomains in State 2, a multibody refinement was performed focusing on the SSU-head, SSU-body, and LSU regions separately. This resulted in resolutions of 3.28Å (SSU-head), 3.04Å (SSU-body), and 2.83Å (LSU) (S2B Fig). The resulting focused refined maps and their half maps were combined using PHENIX [42] to generate a map at 2.9 Å (S2B Fig).

## Model building, refinement, and validation

At the start of this study, no complete and annotated genome was available for *P. locustae*. Hence, to ensure complete coverage of all the relevant ribosomal protein and RNA sequences, we used 3 available, but non-annotated, *P. locustae* genomes [43–45] to create a local database. This database was used to identify *P. locustae* ribosomal protein and rRNA sequences (S2 Table). Yeast or *V. necatrix* protein and rRNA sequences were used with tblastn [46] or blastn against the in-house genome database to identify the respective homologs in *P. locustae*. The high-resolution crystal structure of the yeast ribosome (PDB 4V88 [19]) was used as initial template for modeling the *P. locustae* ribosome in Coot [47]. Lso2 was built de novo in Coot. In the overall structure, a small number of surface-exposed cysteines showed additional density close to the thiol groups, indicating a low level of oxidation. Model refinement was performed against the combined map of State 2 (2.9 Å) with PHENIX [42] using phenix.real_space_refine (version 1.14–3260) and manually defined zinc coordination and rRNA restraints. Model validation was performed according to [48]. All atomic coordinates were randomly displaced by 0.5 Å, followed by refinement against half map 1. The Fourier shell correlation coefficient of the resulting refined model and half map 1 or half map 2 was calculated to evaluate the model for overfitting. Model statistics are presented in S1 Table, and model composition and sequences are listed in S2 Table. While this paper was in preparation, the cryo-EM structures of Lso2 bound to the yeast ribosome [49] and CCDC124 bound to the human ribosome [49,50] were published. These studies confirm the overall structural fold and binding mode of Lso2 described here.

## Map and model visualization

Structure analysis and figure preparation were performed using PyMOL (Schrödinger) [51] and UCSF ChimeraX [52].

## Supporting information

**S1 Fig. Cryo-EM data collection and processing scheme.** (A) Representative cryo-EM micrograph of the microsporidian ribosome. (B) The 5,332 collected micrographs were manually inspected to remove those with drift, poor CTF fits, or low-quality ice, resulting in a total of 5,274 micrographs. Particles were picked using crYOLO [39] and subjected to 1 round of 2D classification (representative 2D class averages shown) in RELION-3.1 [41] to remove picking contaminations. The initial model was generated using cryoSPARC [40]. A consensus refinement yielded a map at 3.0-Å resolution, which was improved further by per-particle CTF refinement to a resolution of 2.73 Å. To isolate the most populated conformation of the dynamic SSU-head region, a focused 3D classification was performed without image alignment. The resulting 3 classes of the SSU-head domain (different shades of yellow) are shown superimposed with the full consensus refined ribosome. Lso2 is highlighted in red. The class with the best resolved SSU-head, Class 2, contained additional density for an E-site tRNA (sky blue), and was refined to an overall resolution of 2.93 Å. Multibody refinement yielded maps with resolutions of 3.28 Å for the SSU-head (EMD-11437-additional map 1), 3.04 Å for the SSU-body (EMD-11437-additional map 2), and 2.83 Å for the LSU (EMD-11437-additional map 3). These maps were combined using PHENIX combine-focused-maps (EMD-11437). *The inset depicts a superposition of Class 1 and 2 to visualize the 2 conformational states of the SSU-head. °To estimate the percentage of ribosomes bound to Lso2, a mask enclosing this region was used for a 3D classification without image alignment. Class 1 shows clear density for Lso2, suggesting that 91.8% of all ribosomes are bound by this factor.
(TIF)

**S2 Fig. Global and local resolution estimation, model validation, and visualization of the model-density fit.** (A–C) Fourier shell correlation (FSC) curves of the consensus refined state 2 (A), the multibody refined maps and the combined final volume (B), and map-to-model cross-validation (C). The thin dashed line indicates an FSC value at 0.143 or 0.5. Curves were obtained from RELION-3.1 [41] (A and B) or EMAN2 [53] (C). (D) The final focused refined map (EMD-11437) is shown (left) next to a core-region cross-section (middle). Densities for eL20, uL6, and the bound nucleotide (highlighted in lime) and Lso2 (right) are displayed in isolation. All maps are colored according to local resolution. Local resolution was estimated using RELION-3.1 and visualized in UCSF ChimeraX [52]. (E) Selected representative cryo-EM densities superimposed with the corresponding models (PDB 6ZU5), colored in blue (LSU), yellow (SSU), or red (Lso2).
(TIF)

**S3 Fig. Conservation of Lso2 in eukaryotes and its ribosome interaction surfaces.** (A) A multiple sequence alignment of Lso2 from microsporidia and selected eukaryotes. The related *S. cerevisae* Lso1 is also included. The C-terminal ends of *M. daphniae* and *Homo sapiens* have been truncated. The domain architecture of Lso2 is presented on the top. (B) Lso2 shown in isolation with side-chains as spheres, colored according to conservation from white (variable) to red (conserved). The conservation was calculated with Homolmapper [54], using the microsporidian sequences shown in (A). (C) Lso2–ribosome interaction interfaces (shades of green) were obtained using EBI PISA [55]. (B and C) Molecular models are shown from PDB 6ZU5.
(TIF)

**S4 Fig. Stepwise reduction of rRNA elements in microsporidia.** (A) LSU region around the polypeptide exit tunnel, shown for *S. cerevisiae* (PDB 4V88 [19]), *P. locustae* (PDB 6ZU5, solved here), and *V. necatrix* (PDB 6RM3 [9]). Eukaryotic ESs and rRNA helices diminish from left to right. Peptide exit tunnels are denoted by a red square. (B) Reduction of the SSU

ESs es6 and es3. *P. locustae* again represents an intermediate state in this reduction.
(TIF)

**S1 Table. Cryo-EM data collection, refinement, and model statistics.**
(PDF)

**S2 Table. Model composition and sequence information.** Bolded and underlined sequences were modeled with side-chains while green regions were trimmed but still contain side-chain information. Sections indicated in yellow were modeled with poly-alanine structural elements, and the ubiquitin moiety of eL40 is indicated in blue.
(PDF)

## Acknowledgments

We thank M. Hall and T. Heidler for their support with data collection at the Umeå Core Facility for Electron Microscopy, and all members of the Barandun laboratory for discussions and critical reading of this manuscript. The electron microscopy data were collected at the Umeå Core Facility for Electron Microscopy, a node of the Cryo-EM Swedish National Facility, funded by the Knut and Alice Wallenberg Foundation, Erling-Persson Family Foundation, Kempe Foundation, SciLifeLab, Stockholm University, and Umeå University.

## Author Contributions

**Conceptualization:** Charles R. Vossbrinck, Jonas Barandun.

**Data curation:** Kai Ehrenbolger, Nathan Jespersen, Himanshu Sharma, Jonas Barandun.

**Formal analysis:** Kai Ehrenbolger, Nathan Jespersen, Himanshu Sharma, Jonas Barandun.

**Funding acquisition:** Jonas Barandun.

**Investigation:** Kai Ehrenbolger, Nathan Jespersen, Himanshu Sharma, Charles R. Vossbrinck, Jonas Barandun.

**Methodology:** Kai Ehrenbolger, Nathan Jespersen, Himanshu Sharma, Charles R. Vossbrinck, Jonas Barandun.

**Project administration:** Jonas Barandun.

**Resources:** Yuliya Y. Sokolova, Yuri S. Tokarev, Charles R. Vossbrinck, Jonas Barandun.

**Supervision:** Jonas Barandun.

**Validation:** Kai Ehrenbolger, Nathan Jespersen, Himanshu Sharma, Jonas Barandun.

**Visualization:** Kai Ehrenbolger, Nathan Jespersen, Himanshu Sharma, Jonas Barandun.

**Writing – original draft:** Kai Ehrenbolger, Nathan Jespersen, Himanshu Sharma, Jonas Barandun.

**Writing – review & editing:** Kai Ehrenbolger, Nathan Jespersen, Himanshu Sharma, Yuliya Y. Sokolova, Yuri S. Tokarev, Charles R. Vossbrinck, Jonas Barandun.

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
