## [Editor Report · Decision Letter 0]

11 Aug 2020

Dear Dr Barandun, 

Thank you for submitting your manuscript entitled "Differences in structure and hibernation mechanism highlight diversification of the microsporidian ribosome." for consideration as a Short Report by PLOS Biology.

Your manuscript has now been evaluated by the PLOS Biology editorial staff, as well as by an academic editor with relevant expertise, and I'm writing to let you know that we would like to send your submission out for external peer review.

Please re-submit your manuscript within two working days, i.e. by Aug 13 2020 11:59PM.

Kind regards,

Roli Roberts

Senior Editor

PLOS Biology

---

## [Decision Letter · Decision Letter 1]

29 Sep 2020

Dear Dr Barandun,

Thank you very much for submitting your manuscript "Differences in structure and hibernation mechanism highlight diversification of the microsporidian ribosome." for consideration as a Short Report by PLOS Biology. As with all papers reviewed by the journal, yours was evaluated by the PLOS Biology editors as well as by an Academic Editor with relevant expertise and by three independent reviewers.

Based on the reviews, we will probably accept this manuscript for publication, assuming that you will modify the manuscript to address the remaining points raised by the reviewers. Please also make sure to address the data and other policy-related requests noted at the end of this email.

IMPORTANT:

a) Reviewer #2 requests that you reproduce (in microsporidia) the in vitro splitting assay performed in the Wells et al study. While this would clearly strengthen your paper, after discussion with the Academic Editor, we will not require it for a Short Report publication in our journal.

b) The Academic Editor asked us to emphasise the need to interpret the bound nucleotide more cautiously (as indicated by rev #3), and would like to see a more direct statement about Lso2 occupancy and the different conformational states.

c) Please address all other concerns raised by the reviewers.

d) Please cite the relevant EDMD and PBD accession numbers in all main and supplementary Figure legends.

We expect to receive your revised manuscript within two weeks. Your revisions should address the specific points made by each reviewer. In addition to the remaining revisions and before we will be able to formally accept your manuscript and consider it "in press", we also need to ensure that your article conforms to our guidelines. A member of our team will be in touch shortly with a set of requests. As we can't proceed until these requirements are met, your swift response will help prevent delays to publication.

- a cover letter that should detail your responses to any editorial requests, if applicable

*Copyediting*

*Published Peer Review History*

*Early Version*

Sincerely,

Roli Roberts

Senior Editor,

rroberts@plos.org,

PLOS Biology

REVIEWERS' COMMENTS:

Reviewer #1:

[identifies himself as Alexey Amunts]

The cryo-EM study by Ehrenbolger and coworkers presents a single-particle structural model of hibernating ribosome from microsporidium pathogen family organism Paranosema locustae in complex with a recycling factor Lso2. Although microsporidian ribosomes generally have reduced rRNA, correlated with minimal known eukaryotic DNA, here the authors have chosen species with a less reduced variant, and remarkably identified an expansion segment reduced to a single nucleotide. From the high-resolution structural models, the authors assigned a function for this nucleotide and could explain why it retained in evolution. The central aspect of the work is the factor Lso2 that blocks catalytic sites. 

The technical work is well executed, the analysis conducted with attention to details and well presented. The comparison to other characterised and more familiar ribosomal complexes is interesting and helpful.

I only have minor comments:

- Figure 4 and related text: perhaps indicate the local resolution for the nucleotide density to assure readers.

- Methods: clarify why three P. locustae genomes were needed.

Reviewer #2:

This manuscript by Ehrenbolger and colleagues replicates a previously published study by Beckmann's group in which Wells and colleagues described hibernation and reactivation of 80S ribosomes with protein Lso2. Because PLoS Biology publishes "scooped" work as a means of improving reproducibility, I request the authors to replicate the most exciting and key finding of Wells and colleagues' study: their finding that Lso2 is not simply an additional hibernation factor, but a factor that helps reactivate hibernating ribosomes by predisposing them for Dom34-mediated splitting into subunits. Otherwise, I find the study by Ehrenbolger and colleagues too incremental and superficial to warrant its publication in PLoS Biology. 

Specifically, I request the authors to express microsporidian homologs of Dom34-recycling system and repeat the in vitro splitting assay shown in Figure 5 of Beckmann's group study (https://journals.plos.org/plosbiology/article?id=10.1371/journal.pbio.3000780).

Also, to extend their analysis, the authors could possibly speculate in their Discussion about why Lso2 does not bind ribosomes during stages of active growth? Does Lso2 possess any serine/threonine residues (at the rRNA/protein interface) which could undergo phosphorylation in response to growth stimuli?

As a minor comment, I disagree with the authors's statement that their study highlights "diversification of the microsporidian ribosome". In my opinion, their study highlights diversification and redundancy of ribosome hibernation mechanisms. For instance, yeast S. cerevisiae can hibernate their 80S ribosomes by using either protein Lso2 or Myb protein 1. But it does not mean that S. cerevisiae 80S ribosomes have two distinct and diversified populations for each of these hibernation factors.

Overall, this study is stunningly illustrated and written in a very articulate and simple manner, and I am sure the authors will be able to easily address my comments. 

Reviewer #3:

This manuscript reports the structure of an inactive microsporidian (P. locustae) ribosome bound by the ribosome hibernation factor Lso2. The ribosome structure presented here shows an intermediate level of rRNA reduction relative to the recently published V. necatrix ribosome. The Lso2 hibernation factor also contrasts with the MDF1 and MDF2 hibernation factors in the V. necatrix ribosome, and allows direct comparison with Lso2 / CCDC124 in other eukaryotes. It is also reported that the structure contains a bound nucleotide between uL6 and uL20. This nucleotide is interpreted to serve as a replacement for an eukaryotic expansion segment that has been reduced in P. locustae.

The structure of this microsporidian ribosome contributes to our understanding of the translational machinery, based on its phylogenetic position and reduced ribosome size. The Lso2 structure is likewise valuable and is complemented by functional studies in yeast (Wang et al., 2018). However, it seems that the presence of the bound nucleotide is interpreted quite strongly in the absence of corroborating data. The proposed model is plausible, but the nucleotide might bind adventitiously, or only in the hibernating state. Gathering the sort of genetic or biochemical evidence that could support this claim is beyond the scope of revisions. However, it is important that the discussion of this structural observation acknowledge these limitations.

---

## [Editor Report · Decision Letter 2]

22 Oct 2020

Dear Dr Barandun,

On behalf of my colleagues and the Academic Editor, Jamie H.D. Cate, I am pleased to inform you that we will be delighted to publish your Short Reports in PLOS Biology. 

PRODUCTION PROCESS

Before publication you will see the copyedited word document (within 5 business days) and a PDF proof shortly after that. The copyeditor will be in touch shortly before sending you the copyedited Word document. We will make some revisions at copyediting stage to conform to our general style, and for clarification. When you receive this version you should check and revise it very carefully, including figures, tables, references, and supporting information, because corrections at the next stage (proofs) will be strictly limited to (1) errors in author names or affiliations, (2) errors of scientific fact that would cause misunderstandings to readers, and (3) printer's (introduced) errors. Please return the copyedited file within 2 business days in order to ensure timely delivery of the PDF proof. 

If you are likely to be away when either this document or the proof is sent, please ensure we have contact information of a second person, as we will need you to respond quickly at each point. Given the disruptions resulting from the ongoing COVID-19 pandemic, there may be delays in the production process. We apologise in advance for any inconvenience caused and will do our best to minimize impact as far as possible.

EARLY VERSION

PRESS 

Kind regards,

Erin O'Loughlin

Publishing Editor, 

PLOS Biology

on behalf of

Roland Roberts,

Senior Editor

PLOS Biology